



# Vertical profiles of global tropospheric nitrogen dioxide (NO₂) obtained by cloud-slicing TROPOMI

Rebekah P. Horner[1], Eloise A. Marais[1], Nana Wei[1], Robert G. Ryan[1,*], Viral Shah[2,3]

[1]Department of Geography, University College London, London, UK
[2]Global Modeling and Assimilation Office (GMAO), NASA Goddard Space Flight Center, Greenbelt, MD 20770, USA
[3]Science Systems and Applications, Inc., Lanham, MD 20706, USA
[*]Now at: School of Geography, Earth and Atmospheric Sciences, University of Melbourne, Melbourne, Australia

*Correspondence to*: Eloise A. Marais (e.marais@ucl.ac.uk); Rebekah P. Horner (rebekah.horner.20@ucl.ac.uk)

**Abstract.** Routine observations of the vertical distribution of tropospheric nitrogen oxides ($NO_x \equiv NO + NO_2$) are severely lacking, despite the large influence of $NO_x$ on climate, air quality, and atmospheric oxidants. Here we derive vertical profiles of global seasonal mean tropospheric $NO_2$ by applying the cloud-slicing method to TROPOspheric Monitoring Instrument (TROPOMI) columns of $NO_2$ retrieved above optically thick clouds. The resultant $NO_2$ are at a horizontal resolution of 1° × 1° for multiple years (June 2018 to May 2022) covering 5 layers in the upper (180-320 hPa and 320-450 hPa) and mid (450-600 hPa and 600-800 hPa) troposphere, and the marine boundary layer (800 hPa to the Earth's surface). Terrestrial boundary layer $NO_2$ are obtained as the difference between TROPOMI tropospheric columns and the integrated column of cloud-sliced $NO_2$ in all layers above the boundary layer. Cloud-slicing $NO_2$ is typically 20-60 pptv throughout the free troposphere and spatial coverage ranges from > 60% in the mid-troposphere to < 20% in the upper troposphere and boundary layer. Our product is similar (within 10-15 pptv) to $NO_2$ data from NASA DC-8 aircraft campaigns (INTEX-A, ARCTAS, SEAC⁴RS, ATom) when both datasets are abundant and sampling coverage is commensurate, but such instances are rare. We use the cloud-sliced $NO_2$ to critique current knowledge of the vertical distribution of global $NO_2$, as simulated with the GEOS-Chem chemical transport model updated to include peroxypropionyl nitrate (PPN) and aerosol nitrate photolysis that liberate $NO_2$ in the lower and mid-troposphere for aerosol nitrate photolysis and upper troposphere for PPN. Multiyear GEOS-Chem and cloud-sliced means are compared to mitigate the influence of interannual variability. We find that for cloud-sliced $NO_2$ the interannual variability is ~10 pptv over remote areas and ~25 pptv over areas influenced by lightning and surface sources. The model consistently underestimates $NO_2$ across the remote marine troposphere by ~15 pptv. In the northern midlatitudes, GEOS-Chem overestimates mid-tropospheric $NO_2$ by 20-50 pptv, as $NO_x$ production per lightning flash is parameterised to be almost double the rest of the world. There is a critical need for in-situ $NO_2$ measurements in the tropical terrestrial troposphere to evaluate cloud-sliced $NO_2$ there. The model and cloud-sliced $NO_2$ discrepancies identified here need to be investigated further to ensure confident use of models to understand and interpret factors affecting the global distribution of tropospheric $NO_x$, ozone and other oxidants.



## 1 Introduction

In the troposphere, nitrogen oxides ($NO_x \equiv NO + NO_2$) influence the formation of tropospheric ozone ($O_3$), a greenhouse gas, and the hydroxyl radical (OH), the main atmospheric oxidant (Atkinson, 2000; Bloss et al., 2005). Due to its influence on OH,

$NO_x$ also indirectly affects the lifetime and abundance of the potent greenhouse gas methane (Wild et al., 2001) and non-methane volatile organic compounds that contribute to $O_3$ and particulate matter pollution (Crutzen and Andreae, 1990; Karl et al., 2007; Marais et al., 2016). $NO_x$ is directly emitted from high-temperature combustion of fossil fuels, from open and domestic burning of biomass, and from natural processes such as lightning and bacteria in soils (Dignon, 1992; Pickering et al., 1998; Jain et al., 2006; Vinken et al., 2014). $NO_x$ also enters the upper layers of the troposphere via downwelling from the

stratosphere (Poulida et al., 1996). The distribution of $NO_x$ varies throughout the troposphere as a result of these sources and due to recycling of $NO_x$ via oxidation, photolysis and thermal decomposition of gas- and aerosol-phase reservoirs of nitrogen (Chatfield, 1994; Moxim et al., 1996; Kotamarthi et al., 2001; Scharko et al., 2014). In the warm lower troposphere where anthropogenic sources dominate, the lifetime of $NO_x$ is a few hours. This increases with altitude to several days in the cold, dry upper troposphere where $NO_x$ is present mostly as NO (Travis et al., 2016), reservoir compounds dominate, and terminal

loss of $NO_x$ via wet deposition in the form of nitric acid ($HNO_3$) is limited (Jaeglé et al., 1998).

Knowledge of the vertical distribution of tropospheric $NO_x$ has been largely informed by in-situ instruments on research and commercial aircraft (Crawford et al., 1996; Brenninkmeijer et al., 1999; Bradshaw et al., 2000; Emmons et al., 2000; Petzold et al., 2015; Stratmann et al., 2016). These aircraft campaigns are few in time and space. The instruments used to measure $NO_2$

are also susceptible to interference from decomposition of thermally unstable reservoir compounds of $NO_x$ (Bradshaw et al., 2000; Browne et al., 2011; Reed et al., 2016). This interference is most severe in the upper troposphere and in remote marine regions where thermally labile $NO_x$ reservoir compounds are abundant and decomposition of these compounds is promoted by the warm instrument inlet (Murphy et al., 2004; Nault et al., 2015; Shah et al., 2023). Studies now supplement these measurements with calculated daytime $NO_2$ concentrations, as NO and $NO_2$ can be assumed to be in photochemical steady

state (PSS) (Davis et al., 1993; Crawford et al., 1996).

Networks of ground-based remote sensing instruments such as Multi Axis Differential Optical Absorption Spectroscopy (MAX-DOAS) and direct-sun Pandora instruments have expanded globally. Still, geographic coverage for both is mostly in the northern hemisphere (Verhoelst et al., 2021). For Pandora, only the total tropospheric column can be derived from total

atmospheric column measurements (Pinardi et al., 2020). MAX-DOAS, under ideal conditions, can retrieve up to four independent layers in the troposphere, though vertical extent at most sites excludes the upper troposphere (Tirpitz et al., 2021). Space-based remote sensing observations used to retrieve vertical column densities (VCDs) of tropospheric $NO_2$ address limited spatial sampling of commercial and research aircraft and Pandora and MAX-DOAS networks by offering daily global coverage, but with only one piece of vertical information in the troposphere (Ryan et al., 2023). These satellite observations



are also impacted by biases in modelled vertical profiles of $NO_2$ required to retrieve VCDs (Verhoelst et al., 2021), in particular in the upper troposphere where satellite observations are most sensitive to tropospheric $NO_2$ (Boersma et al., 2004; Travis et al., 2016; Silvern et al., 2018; Shah et al., 2023).

Mixing ratios of $NO_2$ in distinct layers of the troposphere can be retrieved using so-called cloud-slicing. This technique targets
partial columns (stratospheric + tropospheric) above clouds that are sufficiently optically thick that UV-visible instruments observe discrete layers in the troposphere. Cloud-slicing was first applied by Ziemke et al. (2001) to $O_3$ columns to derive seasonal multi-year mean upper tropospheric $O_3$ mixing ratios in the tropics. Cloud-slicing has since been used to retrieve seasonal mean concentrations of $NO_2$ from the Ozone Monitoring Instrument (OMI) in both the mid (900-650 hPa or 2-4 km) and upper (450-280 hPa or 6-11 km) troposphere at 5° latitude × 8° longitude (500 km × 800 km) as well as in six pressure
levels (centred at 280, 380, 500, 620, 720 and 820 hPa) at 2° × 2° (Belmonte Rivas et al., 2015; Choi et al., 2014; Marais et al., 2018). The OMI cloud-sliced $NO_2$ data provide useful information at very coarse scales (20° × 32°, seasonal) (Marais et al., 2018) and are hindered by large data loss after 2007 when many satellite pixels became obscured by the row anomaly (Torres et al., 2018). More recently, the higher spatial resolution TROPOspheric Monitoring Instrument (TROPOMI) has been used to derive $NO_2$ mixing ratios in the upper troposphere (450-180 hPa or 6-12 km) at finer scales than was possible with
OMI of 1° × 1° (~100 km) (Marais et al., 2021). Cloud-sliced $NO_2$ from TROPOMI has so far only been derived for a single year, as at the time there were frequent updates to the retrieval that led to inconsistencies in the TROPOMI $NO_2$ VCDs used for cloud slicing. TROPOMI $NO_2$ data have since been reprocessed to obtain a consistent data record starting in May 2018.

Evaluation of cloud-sliced $NO_2$ data products is very limited, as coincidence of satellite observations and aircraft campaigns
is rare. Choi et al. (2014) found that the NASA OMI mid tropospheric product is similar (<10% difference) to coincident research aircraft campaign observations, limited to Texas and the Pacific Ocean west of North America. Marais et al. (2021) intercompared seasonal mean cloud-sliced upper tropospheric $NO_2$ from TROPOMI and the NASA OMI product to identify that TROPOMI background values routinely exceed OMI by 12-26 pptv. Given these product disparities, independent evaluation of cloud-sliced $NO_2$ mixing ratios is crucial. Past (2006-2013) NASA DC-8 aircraft campaigns and the more recent
(2016-2018) NASA DC-8 Atmospheric Tomography Mission (ATom) measurement campaign sampled the troposphere from close to the surface to the upper layers of the troposphere, offering the opportunity to evaluate cloud-sliced $NO_2$ mixing ratios over the remote Pacific and Atlantic Oceans (ATom) (Thompson et al., 2022), the Canadian Arctic during the Arctic Research of the Composition of the Troposphere from Aircraft and Satellites (ARCTAS) campaign (Jacob et al., 2010), and the eastern US during the Intercontinental Chemical Transport Experiment – North America Phase A (INTEX-A) (Singh et al., 2006) and
the Studies of Emissions and Atmospheric Composition, Clouds and Climate Coupling by Regional Surveys (SEAC[4]RS) (Toon et al., 2016) campaigns.



Here we derive a global dataset of 4 years of seasonal multiyear mean concentrations of NO$_2$ in five discrete vertical layers in
the troposphere from the planetary boundary layer to the upper troposphere. We evaluate our dataset against directly measured
and calculated (PSS) NO$_2$ from multiple NASA DC-8 aircraft campaigns and go on to use the cloud-sliced data to assess
current understanding of the global vertical distribution of tropospheric NO$_x$ as simulated by the GEOS-Chem chemical
transport model.

## 2 Methods

### 2.1 Cloud-slicing TROPOMI NO$_2$ columns

TROPOMI was launched in October 2017 aboard the Sentinel-5P satellite. The initial TROPOMI nadir spatial resolution of
7.2 km × 3.5 km was enhanced to 5.6 km × 3.5 km in August 2019 (Liu et al., 2021). The swath width is 2600 km, resulting
in daily global coverage at an equator crossing time of 13:30 local solar time (LST). To derive our cloud-sliced product, we
use TROPOMI Level 2 swaths retrieved using a consistent algorithm (version 2.3.1). Data are available as the reprocessed
(PAL) product from 1 June 2018 to 14 November 2021 (https://data-portal.s5p-pal.com/, last accessed 17$^{th}$ February 2022)
and as the offline (OFFL) product from 14 November 2021 to 31 May 2022 (https://s5phub.copernicus.eu/dhus/#/home, last
accessed 7 July 2022; now available at https://dataspace.copernicus.eu/browser/). The cloud-slicing approach was first applied
to TROPOMI by Marais et al. (2021) to derive NO$_2$ mixing ratios in the upper troposphere over a broad pressure range from
450 to 180 hPa. We apply this cloud-slicing, with updates detailed below, to the whole troposphere to derive vertical profiles
of seasonal mean NO$_2$ at the same 1° × 1° resolution as Marais et al. (2021) for multiple years (2018-2022) over five pressure
ranges: one in the boundary layer below 800 hPa (< ~2 km), two in the mid-troposphere at 800-600 hPa (~2-4 km) and 600-
450 hPa (~4-6 km), and two in the upper troposphere at 450-320 hPa (~6-9 km) and 320-180 hPa (~9-12 km).

The first application of cloud-slicing to TROPOMI NO$_2$ is described in detail in Marais et al. (2021). We mostly follow the
same approach. That is, pixels of individual swaths are filtered to isolate observations above optically thick clouds (cloud
radiance fraction > 0.7). These are binned into cloud-top pressures within the 5 targeted pressure ranges on a fixed 1° × 1°
grid. The stratospheric component of the total VCDs is corrected for a 13% underestimate in variance identified by Marais et
al. (2021) from comparison to ground-based direct sun photometer Pandora measurements at the high-altitude (4.2 km) Mauna
Loa site. The corrected stratospheric VCDs are multiplied by the reported stratospheric air mass factors (AMFs) to calculate
stratospheric slant columns. The stratospheric slant columns are then subtracted from the total slant columns to estimate
tropospheric slant columns that are converted to tropospheric VCDs using a geometric AMF. Only clusters of total above-
cloud VCDs with a relatively uniform stratosphere are retained for cloud-slicing. These are identified as clusters of 1° × 1°
pixels with a stratospheric column relative standard deviation < 0.02. A uniform stratosphere ensures that variability in partial
NO$_2$ columns above optically thick clouds is dominated by variability in the troposphere. Cloud-slicing also requires that each



cluster include a representative range of cloud-top pressures (Choi et al., 2014). To ensure this is achieved, we remove clusters
with cloud pressure ranges that are < 60% of the pressure range of each layer (for example, 120 hPa threshold for the 800-600
hPa layer) and that have a large standard deviation (≥ 30 hPa), informed by thresholds used by Choi et al. (2014) and Marais
et al. (2018, 2021).

Next, we regress cloud-top pressures against above-cloud $NO_2$ VCDs for clusters with at least 10 satellite pixels. We replace
the reduced major axis (RMA) regression fit originally used by Marais et al. (2021) with Theil regression, as this reduces
influence from outliers and is better suited to data that are not always normally distributed (Theil, 1950; Sen, 1968). The
regression slope in molecules $cm^{-2}$ $hPa^{-1}$ is converted to $NO_2$ volume mixing ratios in pptv as in Choi et al. (2014). The updated
Theil regression fit addresses the 12-26 pptv overestimate in background values of cloud-sliced upper tropospheric $NO_2$
identified by Marais et al. (2021) from comparison to the OMI upper tropospheric product. It also negates the need for the
large TROPOMI free tropospheric column $NO_2$ bias correction that Marais et al. (2021) used to resolve an apparent
overestimate in TROPOMI compared to free tropospheric $NO_2$ columns derived with measurements from Pandora and MAX-
DOAS instruments at the high-altitude Izaña site. We also find that the outlier filter used by Marais et al. (2021) for cloud-
sliced $NO_2$ > 200 pptv is no longer needed, as it has negligible impact on seasonal mean cloud-sliced $NO_2$ using our updated
approach. As an initial assessment, we compare upper tropospheric cloud-sliced $NO_2$ from our updated cloud-sliced approach
to those from Marais et al. (2021). To ensure a consistent comparison, we recompute our updated cloud-sliced $NO_2$ to cover
the same pressure range (450-180 hPa) and time (June 2019 to May 2020) as Marais et al. (2021) and only compare 1° × 1°
grids with 5 or more cloud-sliced data points in each data product.

The use of a geometric AMF to convert slant columns to vertical columns assumes $NO_2$ mixing ratios within each layer are
relatively well mixed. Belmonte Rivas et al. (2015) estimated that the difference between the geometric AMF and an AMF
that accounts for surface reflectivity, the vertical $NO_2$ profile and atmospheric scattering is <10% in all layers, except the
lowest layer in that work of 770-870 hPa. In this lowest layer, equivalent to the top half of the boundary layer in our work, the
difference in AMFs is up to ~30%. The largest differences occur over land where $NO_x$ emissions from sources such as urban
traffic, industry, soils, and open burning of biomass cause an exponential increase in $NO_2$ with pressure, unlike over the oceans
where the $NO_2$ profile is relatively uniform (Schreier et al., 2015; Wang et al., 2019; Kang et al., 2021; Shah et al., 2023).
Given the steep vertical gradient in terrestrial boundary layer $NO_2$, we instead derive $NO_2$ mixing ratios in the lowest layer
over land as the difference between TROPOMI seasonal mean cloud-free $NO_2$ tropospheric columns and free tropospheric
columns obtained by integrating cloud-sliced $NO_2$ over the four layers above the boundary layer (800-180 hPa).

Cloud fraction and cloud-top height data are from the improved Fast Retrieval Scheme for Clouds from the Oxygen A-band
wide (FRESCO) algorithm called FRESCO-wide (Eskes and Eichmann, 2023). FRESCO-wide minimises the difference
between measured and simulated spectra between 757-758, 760-761 and 765-770 nm and is so-called because the third spectral





window is wider than the 765-766 nm window used in the previous FRESCO-S algorithm (Wang et al., 2008; Van Geffen et al., 2022). The cloud-top pressure retrieved with FRESCO-wide corresponds to an altitude ~1 km lower than the physical cloud-top height, as the cloud-top height retrieval assumes clouds are uniform reflective boundaries (Choi et al., 2014; Loyola et al., 2018). Marais et al. (2021) showed that cloud-sliced $NO_2$ is relatively insensitive to the choice of TROPOMI cloud product. Their use of the TROPOMI Retrieval of Cloud Information using Neural Networks Cloud As Layers (ROCINN-CAL) product yielded upper tropospheric $NO_2$ that was only 4-9 pptv more than that from the FRESCO-S product. The small difference results from an extratropical latitude-dependent divergence in cloud-top heights between the two products. The reprocessed TROPOMI $NO_2$ product (v2.3.1) includes data from two cloud retrieval algorithms, FRESCO-wide and $O_2$-$O_2$ cloud (O22CLD). FRESCO-wide is used here, as we find that it yields greater data density than the O22CLD product and differences in $NO_2$ between the two products for coincident grids are small (< 10%). As of August 2023, ROCINN-CAL had not been reprocessed to obtain a consistent record, so is not used.

## 2.2 NASA DC-8 aircraft observations used to evaluate cloud-sliced $NO_2$

We evaluate our cloud-sliced $NO_2$ against NASA DC-8 campaign data. To mitigate interference from decomposition of $NO_x$ reservoir compounds on measured $NO_2$ over remote regions, we calculate PSS $NO_2$ for ATom measurements over remote oceans and for all measurements made in the upper troposphere. The PSS $NO_2$ calculation assumes a dynamic daytime equilibrium between NO and $NO_2$ resulting from the balance between photolysis of $NO_2$ yielding NO and reaction of NO with oxidants regenerating $NO_2$. Silvern et al. (2018) estimated with GEOS-Chem that oxidation of NO in the southeast US upper troposphere was mostly (75%) by $O_3$ followed by the hydroperoxy radical ($HO_2$) (15%). The remaining 10% is due to oxidation by the methyl peroxy radical ($CH_3O_2$) and halogen monoxides. Given dominance of $O_3$ and $HO_2$ and availability of measurements of these for almost all campaigns used, we calculate PSS $NO_2$ as follows:

$$NO_2 = NO \times \left( \frac{k_1[O_3] + k_2[HO_2]}{j_{NO_2}} \right) \tag{1},$$

where $j_{NO_2}$ is the $NO_2$ photolysis frequency (in $s^{-1}$) and $k$ is the rate constant for oxidation of NO by $O_3$ ($k_1$) and by $HO_2$ ($k_2$) (in $cm^3$ molecule$^{-1}$ $s^{-1}$). The square brackets denote concentrations of $O_3$ and $HO_2$ in molecules $cm^{-3}$. NO and $NO_2$ are in pptv. Values of $j_{NO_2}$, NO, $[O_3]$, and $[HO_2]$ are from direct measurements and $k_1$ and $k_2$ are calculated using the temperature-dependent Arrhenius equations documented in the Jet Propulsion Laboratory (JPL) Chemical Kinetics and Photochemical Data publication number 19 (Burkholder et al., 2020). These for cold upper tropospheric temperatures (~220 K) are $k_1 = 1.2 \times 10^{-14}$ $cm^3$ molecule$^{-1}$ $s^{-1}$ and $k_2 = 1.1 \times 10^{-13}$ $cm^3$ molecule$^{-1}$ $s^{-1}$. Only aircraft data obtained between 12:00 and 15:00 LST, 1.5 hours around the TROPOMI overpass time of 13:30 LST, are used, to ensure consistent sampling of the midday atmosphere and that the PSS assumption is valid. We remove aircraft data influenced by stratospheric air, identified as $O_3$/CO > 1.25 mol



mol$^{-1}$. We also only use aircraft NO data to calculate PSS NO$_2$ if the NO measured is double the NO instrument detection limit
of 6 pptv. This ensures measurements used are distinct from background noise in our PSS calculation (Ryerson et al., 2000;
Yang et al., 2023).

NASA DC-8 aircraft campaigns with direct observations of NO$_2$ and observations needed to calculate PSS NO$_2$ include
INTEX-A in summer 2004 over the United States (Singh et al., 2006), ARCTAS in spring and summer 2008 over the Canadian
Arctic (ARCTAS Science Team, 2011), SEAC$^4$RS in summer and autumn 2013 over the southeast US (SEAC4RS Science
Team, 2014), and ATom once in all seasons from 2016 to 2018 following the same pole-to-pole flight path over the Atlantic
and Pacific Oceans (ATom Science Team, 2021). Direct NO$_2$ measurements are from thermal-dissociation laser induced
fluorescence (TD-LIF) (Di Carlo et al., 2013) for INTEX-A and from chemiluminescence (Ryerson et al., 2000) for all other
campaigns. There are other DC-8 aircraft campaigns, such as the Subsonic Assessment Ozone and NO$_x$ Experiment (SONEX)
over the North Atlantic and the Deep Convective Cloud and Chemistry (DC-3) over the eastern US. These are not included in
our comparison, because SONEX has routine influence of stratospheric air (Fuelberg et al., 2000) and because DC-3 targeted
thunderstorms with large concentrations of NO$_x$ from lightning, so is not representative of a standard atmosphere (Singh et al.,
1999; Barth et al., 2015; Nault et al., 2016). Measurements of HO$_2$ are not available for SEAC$^4$RS, so the PSS NO$_2$ calculation
for this campaign uses average upper tropospheric [HO$_2$] from the other three campaigns. We find that INTEX-A
measurements of NO yield median PSS NO$_2$ values at 450-180 hPa that are anomalously large (150-450 pptv) in comparison
to PSS NO$_2$ from SEAC$^4$RS (30-130 pptv), so no upper tropospheric INTEX-A values are used.

**2.3 The GEOS-Chem chemical transport model**

We use GEOS-Chem to evaluate contemporary knowledge of tropospheric NO$_x$ by comparison to our cloud-sliced NO$_2$ vertical
profiles. For this, we use GEOS-Chem version 13.3.4 (https://doi.org/10.5281/zenodo.5764874; accessed 11 May 2022) to
calculate 4-year seasonal mean NO$_2$ covering the same vertical ranges as cloud-sliced NO$_2$. Model years sampled (1 December
2015 to 30 November 2019) are different to those for TROPOMI, due to a lag in availability of emission inventory data. The
model is driven with NASA Modern Era Retrospective analysis for Research and Applications, version 2 (MERRA-2)
reanalysis meteorology at a horizontal resolution of 2° × 2.5° over 47 vertical layers (30-35 in the troposphere) extending to
0.01 hPa.

Global emissions of all anthropogenic sources except aircraft are from the Community Emissions Data System (CEDS) version
2 for 2015 to 2019 (McDuffie et al., 2020). Aircraft emissions of NO$_x$ are from the Aviation Emissions Inventory Code (AEIC)
for 2005 (Stettler et al., 2011). We use offline grid-independent soil NO$_x$ emissions from Weng et al. (2020), the online Global
Fire Emissions Database version 4 with small fires (GFED4s) (van der Werf et al., 2017) inventory for open burning of
biomass, and offline grid-independent lightning NO$_x$ emissions prepared by Meng et al. (2021) using the parameterisation
detailed in Murray et al. (2012).



GEOS-Chem exhibits a known underestimate in tropospheric $NO_2$ over global oceans, as evidenced by past studies (Guo et al., 2023; Travis et al., 2020; Shah et al., 2023). We address this by updating the GEOS-Chem chemical mechanism to include

photolysis of particle-phase nitrates ($pNO_3$) liberating $NO_x$ as $NO_2$ and as the reservoir compound nitrous acid (HONO) followed by its prompt photolysis to form NO (Andersen et al., 2023; Kasibhatla et al., 2018; Romer et al., 2018; Ye et al., 2017). Photolysis of $pNO_3$ is implemented in GEOS-Chem by scaling the photolysis of nitric acid ($HNO_3$) by an enhancement factor (EF). The EF is 100 for coarse-mode $pNO_3$ and is scaled down using the relative molar concentrations of $pNO_3$ and sea salt aerosol as in Shah et al. (2023) for fine-mode $pNO_3$. This increases lower tropospheric (< 6 km) $NO_2$ over the remote

ocean by up to 15 pptv, but has a smaller effect (< 10 pptv increase) above 6 km where $pNO_3$ is much less abundant (Shah et al., 2023). Photolysis of the $NO_x$ reservoir compound peroxypropionyl nitrate (PPN, $C_2H_5C(O)OONO_2$) leading to formation of $NO_2$ occurs in the atmosphere, but is absent in GEOS-Chem. There are no reported laboratory measurements of $NO_2$ quantum yields from PPN. According to the Harwood et al. (2003) laboratory study, PPN absorption cross sections and quantum yields of the nitrate radical ($NO_3$) are within 10% of peroxyacetyl nitrate (PAN, $CH_3C(O)OONO_2$), so we use PAN

quantum yields and cross sections from Burkholder et al. (2020) to represent PPN photolysis in GEOS-Chem.

For consistent comparison of the model to cloud-sliced $NO_2$, GEOS-Chem is sampled around the TROPOMI overpass (12:00-15:00 LST) following a 3-month spin-up from 1 September to 30 November 2015 for chemical initialisation of the 4-year simulation. Tropospheric $NO_2$ in GEOS-Chem is identified using MERRA-2 tropopause heights and additional filtering is

applied to remove stratospheric intrusions ($O_3/CO > 1.25$ mol $mol^{-1}$). All-sky model scenes are sampled. Marais et al. (2021) determined by applying cloud-slicing to synthetic columns of $NO_2$ simulated with GEOS-Chem that the difference between $NO_2$ under very cloudy and all-sky conditions is small (< 17%). The TROPOMI cloud-sliced data are gridded to the GEOS-Chem grid for the comparison and only grid cells with at least 10 cloud-sliced data points are compared.

## 3 Results and Discussion

**3.1 Vertical distribution of tropospheric $NO_2$ from cloud-slicing TROPOMI**

Fig. 1 shows the spatial distribution of cloud-sliced $NO_2$ in the free troposphere in June-August (JJA) 2018-2021 and December-February (DJF) 2018-2022 and Fig. 2 shows boundary-layer $NO_2$ (below 800 hPa) for the same seasons and years obtained with cloud-slicing over the ocean and differencing over land (Sect. 2.1). The percent filled global $1° \times 1°$ grids is similar in both seasons, though with expected seasonal shifts in regions covered, due to seasonality in the location of clouds

associated with convective features such as the Intertropical Convergence Zone (ITCZ) and absence of clouds over regions of persistent subsidence west of southern Africa and South America. Coverage is greatest in the mid-troposphere and least at 320-180 hPa. Percent coverage averaged over JJA and DJF is 63% of grid cells for 600-450 hPa and 68% for 800-600 hPa covering most of the tropics, subtropics, and midlatitudes. Slightly fewer (38%) result at 450-320 hPa, decreasing to 8% at



320-180 hPa. The few grid squares that are filled at this height mostly occur in the tropics, due to the higher tropopause and
greater abundance of optically thick clouds (Wang et al., 1996). In the boundary layer (Fig. 2), a total of ~14% of the grids are
filled, ~11% for direct cloud-slicing and ~3% for differencing. The latter is limited to locations over land with cloud-sliced
NO₂ in the top upper troposphere layer. Per-layer percent grids filled is similar for March-May and September-November.

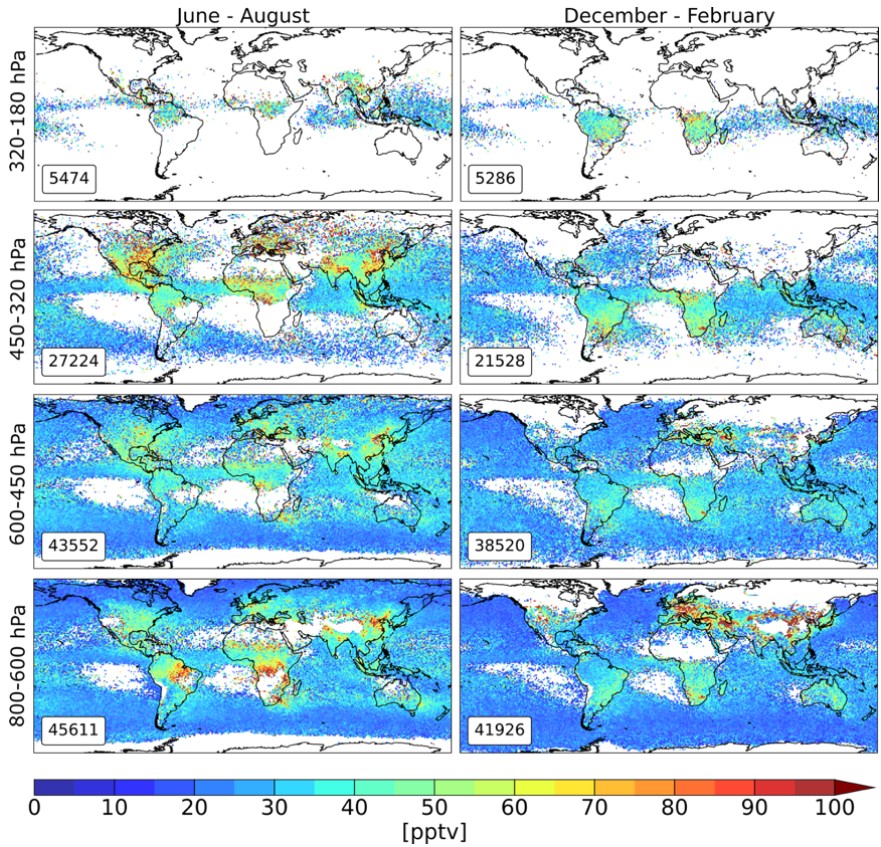

**Figure 1: Seasonal mean NO₂ in the free troposphere obtained by cloud-slicing TROPOMI. Columns are June-August (JJA; left)
and December-February (DJF; right) multiyear (2018-2021 for JJA, 2018-2022 for DJF) means at 1° × 1°. Rows from top to bottom
are 320-180, 450-320, 600-450, and 800-600 hPa. Inset boxes give the number of filled 1° × 1° grids. Boundary layer (below 800 hPa)
data are in Fig. 2.**

Throughout the free troposphere in all seasons (Fig. 1), cloud-sliced NO₂ is typically 20-60 pptv. In the upper troposphere,
lightning NO$_x$ emissions and photolysis of NO$_x$ reservoir compounds sustains NO₂ concentrations of 20-70 pptv over the
oceans and > 90 pptv over the continents in JJA at 450-320 hPa. NO₂ concentrations exceeding 70 pptv in JJA at 450-320 hPa
over North America, China and the Indian subcontinent is due to a combination of lightning and convective uplift of surface
anthropogenic pollution (Bertram et al., 2007; Hudman et al., 2007). NO₂ persists for longer in the cold, dry upper troposphere
(Ehhalt et al., 1992; Jaeglé et al., 1998; Grewe et al., 2001) than in the mid-troposphere below, so NO₂ concentrations are 20



pptv more over Europe and North America at 450-320 hPa than at 600-450 hPa. $NO_2$ over the open oceans is similar (25-50 pptv) throughout the free troposphere and is due mostly to lightning and continental outflow (Kawakami et al., 1997; Zien et al., 2014). $NO_2$ in excess of 55 pptv over South America and 80 pptv over Central Africa at 800-600 hPa results from a mix of intense continental lightning and seasonal open burning of biomass (Andreae et al., 2001; Christian et al., 2003; Duncan et

al., 2003). The burning season in South America starts in July and occurs throughout JJA in southern Africa and throughout DJF in Africa north of the tropics (Van Der Werf et al., 2006; Castellanos et al., 2014; Van Der Velde et al., 2021). $NO_2$ is longer-lived in winter, due to cold conditions and slow photolysis (Dickerson et al., 1982; Kenagy et al., 2018), so over continental Europe large surface sources of anthropogenic $NO_x$ and limited lightning activity especially in comparison to the US contribute to 80 pptv more $NO_2$ in DJF than in JJA at 800-600 hPa.


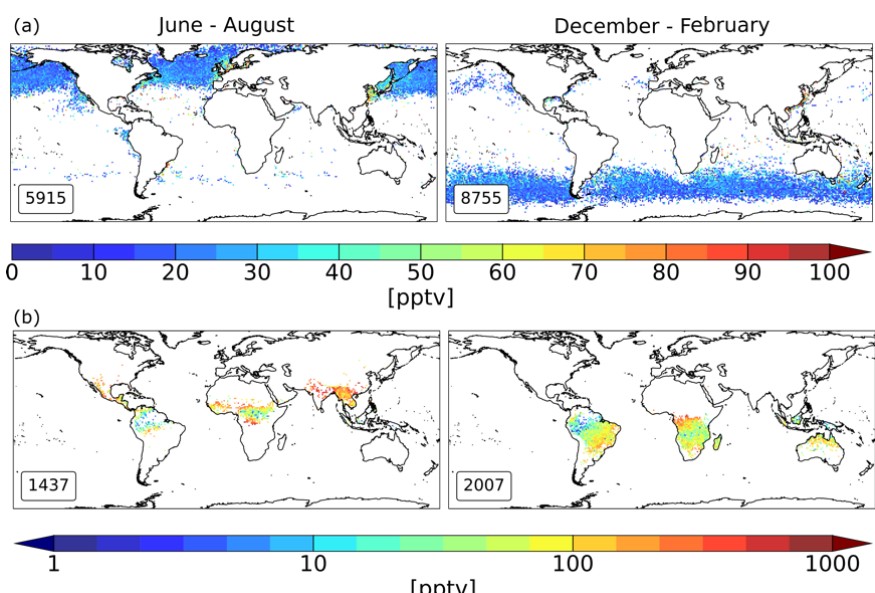

**Figure 2: As in Fig. 1, but for the boundary layer (below 800 hPa). Panels are $NO_2$ from cloud-slicing over the oceans (a) and from the differencing approach over land (b) (see Sect. 2.1 for details). Note colourbar ranges differ in panels (a) and (b), and (b) is on a log scale.**


In the marine boundary layer (Fig. 2 (a)), the typical range in $NO_2$ concentrations is similar to the layers above, except close to coastlines influenced by continental outflow of anthropogenic pollution and local $NO_x$ production from busy harbours. Along the east coast of China, for example, $NO_2$ concentrations are > 90 pptv compared to 25-35 pptv over the remote ocean east of China. In the terrestrial boundary layer (Fig. 2 (b)), $NO_2$ concentrations exceed 30 pptv and peak at 600 pptv over

eastern Brazil in DJF, Central Africa in both seasons and Southeast Asia and the Indo-Gangetic Plain (IGP) in JJA. The peaks in Brazil and Central Africa are due to biomass burning, whereas the peaks in Southeast Asia and the IGP are associated with large urban and industrial sources (Giglio et al., 2010; Ghude et al., 2013; Lu et al., 2024). Steep latitudinal gradients in $NO_2$ of > 100 pptv obtained with the differencing approach for $NO_2$ covering Amazonia and Central Africa is due to influence of



intense seasonal burning of savanna-type vegetation bordering dense tropical forests (Chen et al., 2013; Ossohou et al., 2019;
Jin et al., 2021; Van Der Velde et al., 2021).

The seasonal mean cloud-sliced $NO_2$ at 450-180 hPa obtained by Marais et al. (2021) that we compare to our data for the same
vertical extent and time period (Sect. 2.1) ranges from $NO_2 > 80$ pptv over terrestrial regions to $< 50$ pptv over remote oceans.
The two datasets are spatially consistent in all seasons, yielding Pearson's correlation coefficients (R) of 0.74 in JJA, 0.70 in
SON, 0.64 in DJF and 0.65 in MAM. Marais et al. (2021) $NO_2$ is on average 26% more than we obtain with our updated cloud
slicing. This difference, decomposed into variance and background using RMA regression, is 25-37% more variance and 17-
22 pptv less background $NO_2$ in our data across all four seasons. The greater background values in Marais et al. (2021) are
from susceptibility of their approach to outliers (Sect. 2.1).

We also examine the size of interannual variability (IAV) in tropospheric $NO_2$, according to our cloud-sliced data. This is
shown in Fig. 3 for JJA and DJF for a select year (2021 for JJA, December 2020 to February 2021 for DJF), calculated as the
absolute difference between cloud-sliced $NO_2$ in these years and the multiyear mean (Fig. 1). Only three of the five layers are
shown, as coverage is poor for individual years for the other two layers. IAV data are obtained for <1% of all $1° \times 1°$ grid cells
at 180-320 hPa and just 2% in the boundary layer. IAV in the layers shown in Fig. 3 is typically ~10 pptv over the remote
ocean and ~25 pptv over continental regions (eastern US, Europe, tropics). The greater IAV over the continents is due to
influence of anthropogenic, open biomass burning and lightning $NO_x$ emissions. IAV $NO_2$ is about 20-50% of the variability
the multiyear means in Fig. 1 and 2. Relatively large IAV $NO_2$ over the remote oceans is restricted to the edges of sampled
areas in the subtropics that have low data density, due to the proximity to regions of persistent subsidence where retrievals
from cloud-slicing are not always successful.




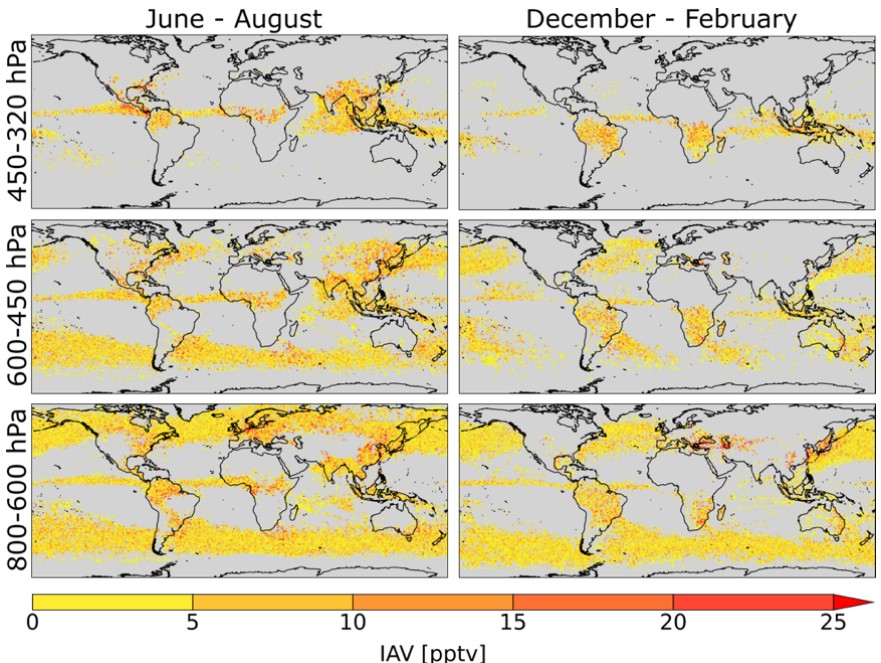

**Figure 3: Free tropospheric NO₂ interannual variability (IAV). Panels are single-year NO₂ IAV obtained as the absolute difference between single-year (left: JJA 2021; right: DJF 2020-2021) and multiyear mean cloud-sliced NO₂ for the 3 layers with greatest geographic coverage. Only grid squares with at least 5 cloud-sliced data points in the single-year means are compared.**


## 3.2 Evaluation of cloud-sliced NO₂ with observed and calculated (PSS) NO₂

Fig. 4 shows the regions selected to intercompare cloud-sliced and DC-8 NO₂ obtained with direct measurements and PSS NO₂ (Section 2.2). Selected regions include the North Atlantic Ocean sampled during ATom, the Canadian Arctic sampled during ARCTAS and ATom, the eastern United States sampled during SEAC⁴RS and INTEX-A, and the Pacific Ocean

sampled during ATom. These regions were chosen to optimise coincidence of aircraft data in all five layers. In many instances, though, coincidence is over a limited extent of the sampling domain, especially the Pacific Ocean in all layers. Domains sampled in all seasons due to the ATom campaign include the Canadian Arctic and the Pacific and Atlantic Oceans. The most sampled time period is JJA, the greatest regional coverage is over the eastern US, and the mid-tropospheric layers (800-600 and 600-450 hPa) have the most DC-8 data. According to the DC-8 NO₂ data, hotspots (NO₂ > 200 pptv) occur over the US

terrestrial boundary layer where there are large surface NO$_x$ emissions. Much lower concentrations of < 25 pptv over the remote ocean are due to absence of large local sources.



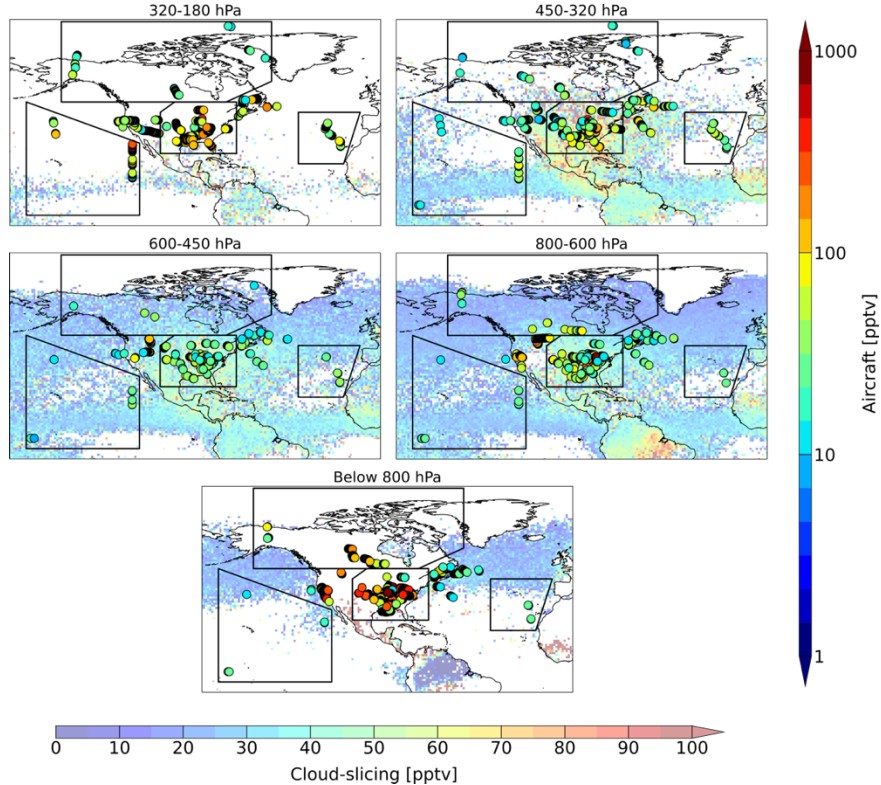


**Figure 4: Maps of tropospheric NO₂ over the north-western hemisphere in June-August for the five cloud-slicing pressure ranges. Filled circles are DC-8 NO₂ data along DC-8 flight tracks (Sect. 2.2). Background values are cloud-sliced NO₂. Polygons show the regions sampled for comparison of aircraft and cloud-sliced NO₂ in Fig. 4 and 5. These are the North Atlantic, the Canadian Arctic, the eastern United States and the Pacific.**


Fig. 5 and 6 compare median DC-8 and cloud-sliced NO₂ concentrations in MAM and JJA (Fig. 5) and SON and DJF (Fig. 6) for the polygons in Fig. 4. Cloud-slicing data are for 2018-2021 in JJA and SON, 2018-2022 in DJF, and 2019-2022 in MAM. JJA data are compared to the ARCTAS, SEAC⁴RS, INTEX-A and ATom-1 campaigns, DJF to ATom-2, SON to ATom-3 and SEAC⁴RS, and MAM to ATom-4 and ARCTAS. Vertical profiles of DC-8 NO₂ are relatively stable (~25-50 pptv) throughout

the troposphere over Pacific and North Atlantic Oceans and increase exponentially to ~75-300 pptv in the boundary layer over the eastern US and the Canadian Arctic. Most cloud-sliced NO₂ in the mid-troposphere and in the 320-450 hPa layer in the upper troposphere are < 10 pptv different to DC-8 NO₂ in the extensively sampled eastern US and < 25 pptv in the other locations for medians obtained with more than 5 data points. The greater variability in the DC-8 data in each layer (larger interquartile ranges), is because DC-8 are single year measurements, whereas cloud-sliced NO₂ are multiyear means.






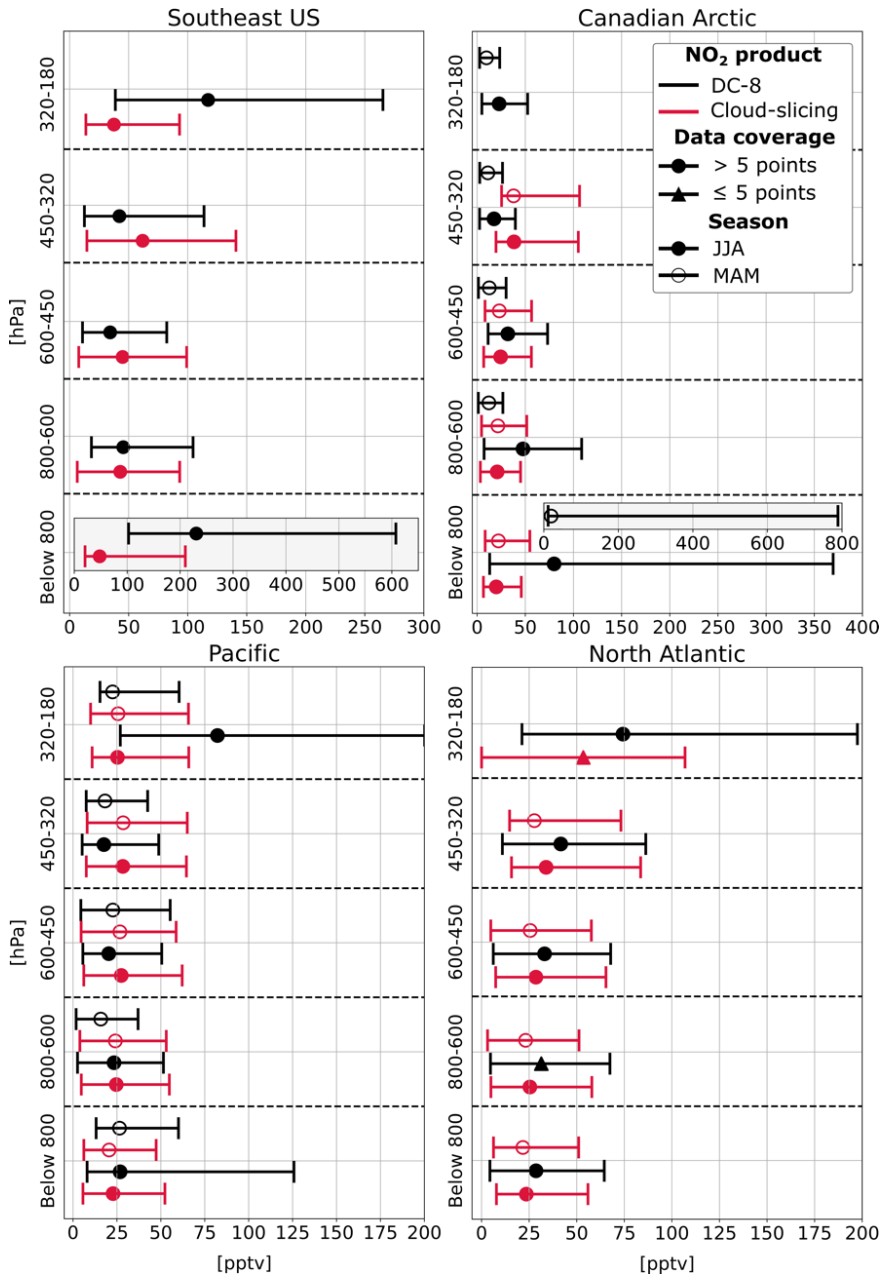

**Figure 5: Comparison of seasonal mean vertical profiles of DC-8 and cloud-sliced tropospheric NO₂. Symbols are median values for the sampling domains in Fig. 3 for data in MAM (open) and JJA (filled). Symbol shapes differentiate medians obtained with £ 5 (triangle) and > 5 (circle) data points in either dataset. Error bars are interquartile ranges (IQR). NO₂ concentration scales differ and inset boxes in the top row show boundary-layer NO₂ exceeding the *x*-axis range.**

Large differences between DC-8 and cloud-sliced NO₂ occur in the boundary layer and the top tropospheric layer. In these layers, there are few coincident data points. Most DC-8 data are over land influenced by ground-based sources like intense biomass burning in the boundary layer (Alvarado et al., 2010; Bian et al., 2013) and lightning and convective uplift of surface



pollution in the upper troposphere, whereas most cloud-sliced $NO_2$ in these 2 layers are over the ocean (Fig. 4). The cluster of points in the boundary layer over New England in the northeast US in Fig. 4 have similar coverage from both datasets. These are on median 30 pptv (IQR: 20-50 pptv) for DC-8 and 25 pptv (IQR: 20-30 pptv) for cloud-sliced $NO_2$. New England is not included in our comparison in Fig. 5 and 6, as sampling over this location is limited to JJA during INTEX-A.


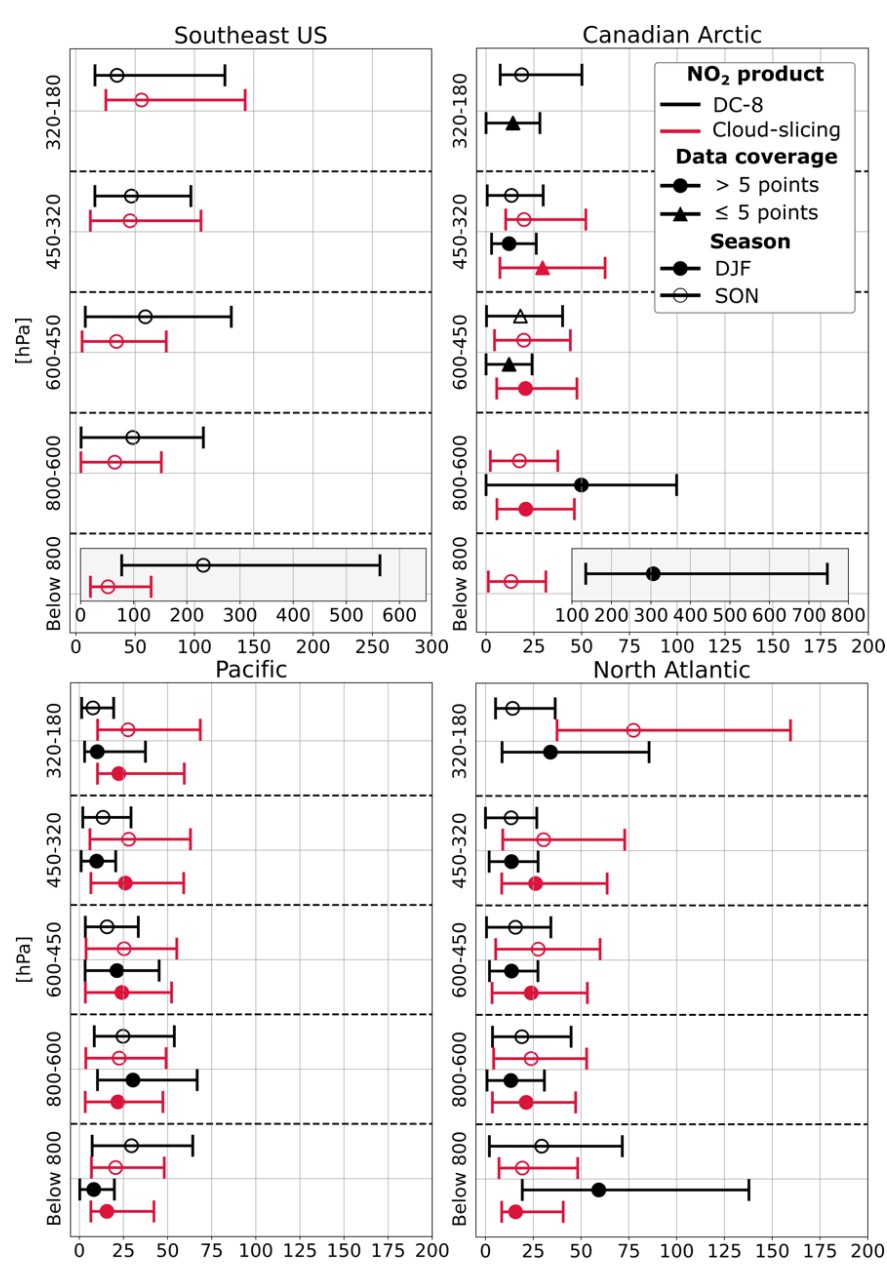

**Figure 6: As in Fig. 4, but for SON (open symbols) and DJF (filled symbols).**





### 3.3 Comparison of cloud-sliced vertical profiles to synthetic GEOS-Chem profiles

Fig. 7 shows the percent difference between multiyear mean GEOS-Chem and cloud-sliced NO₂ for June-August and December-February obtained after regridding the cloud-sliced NO₂ to the GEOS-Chem 2° × 2.5° grid. Multiyear means in both datasets are compared to minimise influence of interannual variability quantified in Section 3.1. In general, GEOS-Chem NO₂ is 30-80% (10-25 pptv) less than cloud-sliced NO₂ in remote locations. Specifically, the Southern Ocean in all layers retrieved, South America throughout the free troposphere, and all grid cells except those over Africa in the upper troposphere.

Similar spatial patterns and magnitudes of discrepancies to those plotted in Fig. 7 occur in March-May and September-November.

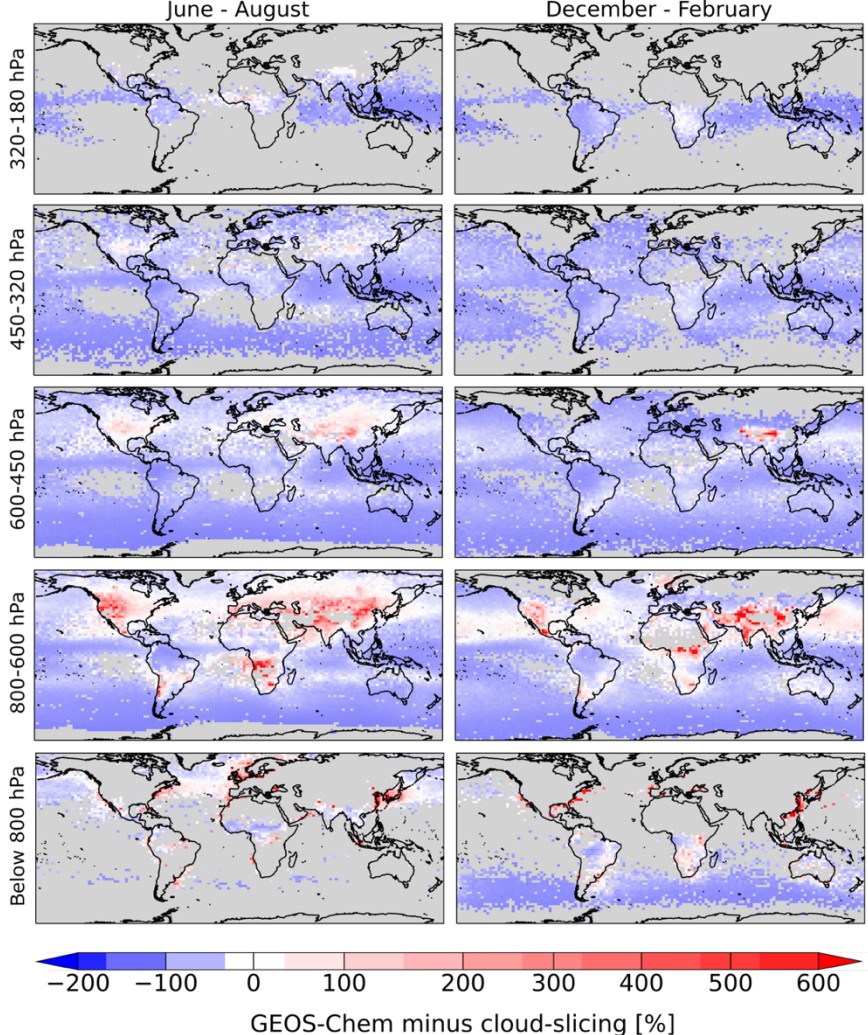



**Figure 7: Percent difference between cloud-sliced and GEOS-Chem vertical profiles of tropospheric NO₂. Maps are at 2° × 2.5°. Blue (red) indicates the model is less (more) than the cloud-sliced NO₂. Percent difference is calculated as ((GEOS-Chem minus cloud-sliced)/cloud-sliced) for all cloud-sliced 1° × 1° grid squares filled in each year sampled.**

Inclusion of nitrate photolysis in GEOS-Chem decreases the model underestimate in $NO_2$ over remote regions from 40-80 pptv to on average ~15 pptv in the mid-troposphere. A relatively large model underestimate of 25-40 pptv over oceans may be due to uncertainties in the enhancement factor used to parameterize nitrate photolysis (Section 2.3) (Shah et al., 2023). PPN photolysis is most effective at increasing $NO_2$ in the 2 layers in the upper troposphere where it is abundant and thermally stable, so photolysis dominates its conversion to $NO_2$. In JJA, for example, PPN photolysis contributes ~65 pptv $NO_2$ over the northern midlatitudes and isolated enhancements of 50-60 pptv over southeast Asia and extending from Mozambique to Madagascar. As a result of PPN photolysis, the discrepancy between the model and cloud-sliced upper tropospheric $NO_2$ is relatively small (10-30 pptv) over the terrestrial northern midlatitudes. The model exceeds the cloud-sliced data by 20-50 pptv over the northern midlatitudes at 600-450 hPa during the summer lightning season north of 35°N. These are the latitudes at which lightning $NO_x$ production rates in GEOS-Chem almost double from 260 moles per flash (mol $fl^{-1}$) to the south to 500 mol $fl^{-1}$ to the north (Murray et al., 2012). The effect of this on $NO_2$ is also evident at 450-320 hPa, though the spatial extent and dataset differences are smaller in this layer. 500 mol $fl^{-1}$ prescribed to northern midlatitude lightning far exceeds observationally constrained global mean estimates of ~280 mol $fl^{-1}$ (Marais et al., 2018) and regional mean estimates of 180 mol $fl^{-1}$ for the northern midlatitudes (Bucsela et al., 2019) and 230-360 mol $fl^{-1}$ for the US and western Atlantic (Allen et al., 2021).

The largest differences between the two datasets occur in the boundary layer along coastlines in North America, Europe and China influenced by anthropogenic pollution. This may in part be due to the different years targeted. COVID lockdowns influenced surface emissions of traffic $NO_x$ in the cloud-sliced data and anthropogenic $NO_x$ emissions are steadily declining over North America, Europe and China as a result of air quality regulation (Zhao et al., 2013; Lloret and Valiela, 2016; Clappier et al., 2021). Both COVID lockdowns and emissions reductions policies would contribute to a model overestimate in $NO_2$. GEOS-Chem also exceeds cloud-sliced $NO_2$ at multiple locations in the 800-600 hPa layer. These include southern Africa in JJA and northern Africa in DJF coincident with the dry burning season of these regions, central Asia in all seasons where there are large sources of anthropogenic pollution. The apparent model overestimate over western US at 600-800 hPa occurs in all seasons and may result from a combination of factors. The TROPOMI sampling period includes the high-fire year (2020) (Albores et al., 2023) and the model does not, affecting the comparison in seasons coincident with the fire season (JJA, SON). The number of cloud-sliced data points are also relatively few over this region of subsidence. It is difficult to diagnose discrepancies in the tropical terrestrial boundary layer, as anthropogenic emissions inventories are prone to misrepresenting sources unique to the tropics (Duncan et al., 2003; Marais and Wiedinmyer, 2016; Vohra et al., 2022) and there are no suitable independent in-situ measurements to validate the differencing approach we use to derive $NO_2$.



## 4 Conclusions

Global vertical profiles of tropospheric $NO_2$ were obtained for five discrete layers (180-320 hPa, 320-450 hPa, 450-600 hPa,
600-800 hPa, and below 800 hPa) by cloud-slicing TROPOMI total columns of $NO_2$ above optically thick clouds. These we
assessed against directly measured and calculated (photostationary steady-state) NASA DC-8 aircraft $NO_2$ measurements from
2004 to 2018. We then applied our cloud-sliced $NO_2$ to evaluate contemporary understanding of climatological tropospheric
$NO_x$ as simulated by GEOS-Chem. We found that coverage from cloud-slicing is greatest in the mid-troposphere (60-70%)
where there is an abundance of optically thick clouds and least (8% coverage, mostly in the tropics) in the upper troposphere.
Cloud-sliced $NO_2$ ranges from < 35 pptv throughout the troposphere over remote marine regions, to 20-60 pptv in the free
troposphere over continents, to 160-380 pptv in the boundary layer over source regions in the US, Europe and Asia. Free
tropospheric $NO_2$ exhibits very little interannual variability, ranging from ~10 pptv over oceans to ~25 pptv over land.

We determined from comparison of cloud-sliced $NO_2$ to NASA DC-8 aircraft observations that cloud-sliced $NO_2$ differs from
DC-8 $NO_2$ by just 5-10 pptv when sampling in both datasets is abundant and consistent. It was not feasible to assess cloud-
sliced $NO_2$ in the boundary layer and in the highest cloud-sliced layer, due to a lack of sufficient coincident data in the tropics.
The GEOS-Chem model that represents contemporary understanding of tropospheric $NO_x$ simulates $NO_2$ that is typically 10-
40 pptv less than cloud-sliced $NO_2$ in the remote upper troposphere and over the remote oceans. This is a substantial
improvement on the > 40 pptv model underestimate before accounting for $NO_x$ recycling in the upper troposphere via PPN
photolysis and in the middle and lower troposphere via aerosol nitrate photolysis. Differences are greater over source regions
influenced by lightning and open burning of biomass and with evolving anthropogenic emissions due to rapid development,
policies and events like lockdowns in response to the COVID-19 pandemic. A model high bias of 50 pptv over the northern
hemisphere mid-troposphere in June-August points to an issue with the model lightning $NO_x$ production rates that are almost
double production rates everywhere else.


Limited coincident reliable observations to validate cloud-sliced $NO_2$ remains a challenge, but as we demonstrate cloud-sliced
$NO_2$ hold value for assessing air quality, chemical transport, and Earth System models to identify differences that warrant
further investigation, especially given reliance on these models to understand complex tropospheric chemistry, inform policies,
and retrieve trace gas abundances from satellites. Geostationary instruments will further enhance the utility of cloud-sliced
$NO_2$ datasets to also investigate daytime variability in vertical profiles of tropospheric $NO_x$.



*Data Availability.* The multiyear seasonal mean $NO_2$ from cloud-slicing TROPOMI and from simulating the GEOS-Chem
model are publicly available from the UCL Data Repository (https://doi.org/10.5522/04/25782336).

*Author contributions.* Study concept by EAM and RPH. RPH led the writing and analysis, simulated GEOS-Chem and cloud-
sliced TROPOMI $NO_2$ with supervision from EAM. NW provided the NASA DC-8 python processing code. RGR updated the
GEOS-Chem model to include PPN photolysis. VS updated the GEOS-Chem model to include particulate nitrate photolysis.
All authors reviewed and edited the manuscript.

*Competing interests.* The authors declare that they have no conflict of interest.

*Acknowledgements.* This research has been supported by the European Research Council under the European Union's Horizon
2020 research and innovation programme (through a Starting Grant awarded to Eloise A. Marais, UpTrop [grant no. 851854]).
We are grateful to NASA DC-8 aircraft campaign teams for access to observations, specifically the $NO_2$ measurement PIs:
Ronald Cohen (INTEX-A), Andrew Weinheimer (ARCTAS), Thomas B. Ryerson (SEAC[4]RS), and Chelsea Thompson
(ATom).



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
