# Peer review of "Vertical profiles of global tropospheric nitrogen dioxide (NO2) obtained by cloud-slicing TROPOMI"

_EGUsphere, 2024_

## Referee Comment (RC1)

**Review:**

**Rebekah P. Horner et al., **Vertical profiles of global tropospheric nitrogen dioxide (NO₂) obtained by cloud-slicing TROPOMI**

**Summary:**

In this study, the authors have used 4 years of data from TROPOMI and applied cloud-slicing to obtain a seasonal climatology of $NO_2$. The study builds on previous cloud-slicing investigations, particularly the work of Marais et al. (2021), but uses an improved algorithm to obtain $NO_2$ climatological profiles in 5 layers, rather than over a single range of pressures. The authors compare their results to modeled $NO_2$ from GEOS-Chem, as well as DC-8 aircraft data from several aircraft campaigns. While some earlier studies were based on OMI data, the present work is the first to apply cloud slicing to higher-resolution TROPOMI measurements and obtain altitude-dependent $NO_2$ mixing ratios. As such, it is an excellent demonstration of how profile information can be obtained from nadir viewing satellites.

The methods described here appear rigorous and the authors clearly explain the algorithmic choices adopted in their approach. I think the paper can be published in nearly its present form. Below are a few minor questions and suggested additions (below).

**Comments:**

(1) Figures 1, 2, 3, 7 show cloud-sliced $NO_2$, its IAV and percentage differences relative to GEOS-Chem at various levels. There are geographic gaps at all levels, 320-180 hPa in particular, which make BL retrievals in these areas impossible. It is difficult to find regions where one can assess how much each level contributes, especially the BL, contributes to the total column. A useful addition would be maps of total column $NO_2$ from cloud slicing, the TROPOMI seasonal cloud-free climatology, and/or GEOS-Chem. Another interesting, but non-essential, addition would be a mean GEOS-Chem profile over an area like the eastern US or a marine region.

(2) The 320-180 hPa cloud-slice data are extremely sparse, if not non-existent, in large geographic areas. Where do the cloud-sliced retrievals in these regions shown in figures 5 and 6 come from? How many such data points are there and why aren't the IQRs larger? Can the number of data points be indicated in the figures as they are for DC-8?

(3) At the end of section 2.2 (page 7), it is stated that no INTEX-A data were used in the upper troposphere, but pages 12 and 13 mention INTEX-A were used in the comparisons. Please add few words to restate that the upper-left panels in figures 5 and 6 do not include these data at 320-180 hPa. Is this also true for 450-320 hPa? A separate question is why no INTEX-B data (e.g. Boersma et al.; 2008) were included. Might their high-altitude measurements be more reliable (in spite of similar instrumentation)?

(4) In figure 4, caption should say "Fig. 5 and 6."

(5)  In figure 5, the caption should say "Fig. 4." and "$\leq 5$".

(6) In figure 6, the caption should refer to "Fig 5."

---

## Author Response (AR1)

**Responses to reviewers**

Ms. Reference Number: EGUSPHERE-2024-1541, doi:10.5194/egusphere-2024-1541

Title: Vertical profiles of global tropospheric nitrogen dioxide ($NO_2$) obtained by cloud-slicing TROPOMI

Journal: Atmos. Chem. Phys.

Point-by-point responses to reviewers are included below. Reviewer comments are in blue. Responses are in black, and line and figure numbers are consistent with the tracked changes manuscript uploaded as a PDF.

**Responses to Anonymous Referee #1:**

*Summary:*

*In this study, the authors have used 4 years of data from TROPOMI and applied cloud-slicing to obtain a seasonal climatology of $NO_2$. The study builds on previous cloud-slicing investigations, particularly the work of Marais et al. (2021), but uses an improved algorithm to obtain $NO_2$ climatological profiles in 5 layers, rather than over a single range of pressures. The authors compare their results to modeled $NO_2$ from GEOS-Chem, as well as DC-8 aircraft data from several aircraft campaigns. While some earlier studies were based on OMI data, the present work is the first to apply cloud slicing to higher-resolution TROPOMI measurements and obtain altitude-dependent $NO_2$ mixing ratios. As such, it is an excellent demonstration of how profile information can be obtained from nadir viewing satellites. The methods described here appear rigorous and the authors clearly explain the algorithmic choices adopted in their approach. I think the paper can be published in nearly its present form. Below are a few minor questions and suggested additions (below).*

*Comments:*
*(1) Figures 1, 2, 3, 7 show cloud-sliced $NO_2$, its IAV and percentage differences relative to GEOS-Chem at various levels. There are geographic gaps at all levels, 320-180 hPa in particular, which make BL retrievals in these areas impossible. It is difficult to find regions where one can assess how much each level contributes, especially the BL, contributes to the total column. A useful addition would be maps of total column $NO_2$ from cloud slicing, the TROPOMI seasonal cloud-free climatology, and/or GEOS-Chem. Another interesting, but*

*non-essential, addition would be a mean GEOS-Chem profile over an area like the eastern US or a marine region.*

The boundary layer $NO_2$ data in Fig. 2(a) are independent of the spatial distribution of the cloud-sliced $NO_2$ in Fig. 1 for the layers above, as these are derived directly with cloud slicing. The boundary layer $NO_2$ in Fig. 2(b) use the sum of the cloud-sliced $NO_2$ from all the other layers (800-180 hPa) and subtracts this from the TROPOMI tropospheric column. This requires there to be data in all layers in the free troposphere, which is reflected in the spatial coverage in the figure. We now include additional text to make this clearer in the Methods (line 162) and the description of Figure 2 (b) (lines 310-311).

To address the comment that it is difficult to assess the contribution from each layer, we include a new figure (Figure 3, pasted below) showing the contribution of each layer to the total tropospheric column for grid cells with coverage in all layers. This limits the assessment to the tropics and subtropics. The contents of the new figure are detailed at lines 333-338.

[Figure]

**Figure 3: Seasonal mean percentage contribution of each cloud-sliced layer to the tropospheric column. Columns are June-August (JJA; left) and December-February (DJF; right). Rows from top to bottom are 320-180, 450-320, 600-450, 800-600, and 1100-800 hPa. Data are multiyear means at 1° × 1°.**

*(2) The 320-180 hPa cloud-slice data are extremely sparse, if not non-existent, in large geographic areas. Where do the cloud-sliced retrievals in these regions shown in figures 5 and 6 come from? How many such data points are there and why aren't the IQRs larger? Can the number of data points be indicated in the figures as they are for DC-8?*

Fig. 5 shows that, though sparse, there are cloud-sliced data points in all polygons except for the Canadian Arctic at the 320-180 hPa pressure range. We do already discuss that coincidence is an issue in comparing DC-8 and cloud-sliced $NO_2$ in the 320-180 hPa layer (line 455). We update the text in this paragraph to point to Fig. 5 (formerly Fig. 4) where the spatial coincidence (or lack thereof) is illustrated. We update Fig. 6 (formerly Fig. 5) caption to make it clearer that the data coverage of no more than or more than 5 data points pertains to both datasets.

*(3) At the end of section 2.2 (page 7), it is stated that no INTEX-A data were used in the upper troposphere, but pages 12 and 13 mention INTEX-A were used in the comparisons. Please add few words to restate that the upper-left panels in figures 5 and 6 do not include these data at 320180 hPa. Is this also true for 450-320 hPa? A separate question is why no INTEX-B data (e.g. Boersma et al.; 2008) were included. Might their high-altitude measurements be more reliable (in spite of similar instrumentation)?*

We update the text in line 222 to state the pressure range over which INTEX-A measurements are not used, so that it is clearer that this includes both the 320-180 hPa and the 450-320 hPa pressure ranges. We also update the text in the Results to remind the reader of this (lines 384-385).

We now include INTEX-B measurements for March-May (MAM) (Figure 6; formerly Figure 5) within the Southeast US and Pacific domains. The updated Fig. 6 is pasted below. The figure caption remains the same. Text has been updated throughout the manuscript to include INTEX-B.

[Figure]

Changed (line 378).

Changed (lines 417-418).

Changed (line 440).

**Responses to Anonymous Referee #2:**

*Summary:*

*This paper describes an extension of previous efforts in total column $NO_2$ cloud slicing in order to obtain vertical profiles of $NO_2$ amount. Building on the work of Marais et al. [2021], who derived $NO_2$ mixing ratios in the upper troposphere from a single year of TROPOMI data, the authors refine the earlier technique in several ways and extend the analysis to the surface, based on 4 years of TROPOMI using a self-consistent retrieval throughout the period. The new cloud-sliced climatology is compared with both aircraft in situ data and GEOS-Chem model output. The analysis presented is very thorough and thoughtfully presented.*

*Comments:*

*(1) In section 2.1, the cloud-slicing technique is nicely explained in the text. The addition of an equation(s), would make it easier to understand how the authors go from a total column amount to mixing ratios in individual layers of the troposphere.*

This conversion equation is already stated by Choi et al. (2014), so we update line 140 to state that this conversion is from Equation (5) of Choi et al. (2014).

*(2) Also in section 2.1, a better of explanation of what "informed by thresholds used by Choi et al." (line 131) means is needed.*

Lines 131-132 state that we use these thresholds to ensure that we have a representative range of cloud top pressures during cloud-slicing. We update text in line 134 to clarify that these thresholds are consistent with cloud-slicing in Choi et al. (2014) and Marais et al. (2018, 2021).

*(3) Please explain what "$NO_2$ mixing ratios within each layer are relatively well mixed" (lines 149–150) means. Do you mean $NO_2$ is well mixed or the mixing ratio is constant within the layer?*

We have rewritten the text in lines 152-153 to clarify that the assumption is that the vertical distribution of NO₂ is relatively constant.

*(4) Better rationales for the thresholds used in the analysis are needed. For example, comparisons between cloud-sliced and GEOS-Chem grids are made only when at least 10 cloud-sliced data points are within the cloud-sliced grid (lines 248–249). Why 10 and not 5 or 15?*

The text in lines 260-262 has been changed to clarify that we use a threshold of 10, as this we find to be optimum for good data coverage and screening for non-representative data.

*(5) Figures 1 and 2: Recommend adding "n=" to the insets in each panel, indicating the number grids used. It will make the figures easier to interpret without reading the text.*

Fig. 1 and Fig. 2 have been modified as below to include "n=" on the inset of each panel and are pasted below. The figure captions remain the same.

[Figure]

[Figure]

*(6) It is hard to derive much quantitative information about the cloud-sliced vs. aircraft $NO_2$ from Figure 4. The maps are valuable in defining the geographic regions analyzed subsequently, and the larger circles show where the aircraft data are nicely. No change is requested—just a comment.*

Thank you for the comment. As you have stated, we show this figure to highlight the spatial distribution and number of data points that we then use for the comparison in the 2 figures that follow, especially to point out layers such as 320-180 hPa with limited spatial overlap between the cloud-sliced and aircraft data to aid in interpreting differences between the two datasets.

*(7) "The greater variability in the DC-8 data in each layer (larger interquartile ranges), is because DC-8 are single year measurements, whereas cloud-sliced $NO_2$ are multiyear means" (lines 353–354): I do not understand why the longer time period would necessarily reduce the IQR. In fact, I might have expected the opposite, as multi-year variability might increase the range of values sampled. Could this instead be due to a difference in the representativeness of the measurement, seeing as the TROPOMI footprints are larger than the DC-8 sampling and thus average out smaller-scale features? In any case, more explanation is needed of this quoted statement.*

In retrospect the variability in IQRs is more nuanced than we initially described. Both data have instances of larger IQRs that typically correspond to fewer data points, as indicated by

Figure 5 (formerly Figure 4). Text is rewritten to reflect this (lines 389-390) and we add text to highlight the limited coincidence in the highest layer in Figure 5 (line 473).

**References:**

Choi et al., 2014, doi: 10.5194/acp-14-10565-2014

Marais et al., 2018, doi: 10.5194/acp-18-17017-2018

Marais et al., 2021, doi: 10.5194/amt-14-2389-2021